# CN133, a Novel Brain-Penetrating Histone Deacetylase Inhibitor, Hampers Tumor Growth in Patient-Derived Pediatric Posterior Fossa Ependymoma Models

**DOI:** 10.3390/cancers12071922

**Published:** 2020-07-16

**Authors:** Roberta Antonelli, Carlos Jiménez, Misha Riley, Tiziana Servidei, Riccardo Riccardi, Aroa Soriano, Josep Roma, Elena Martínez-Saez, Maurizio Martini, Antonio Ruggiero, Lucas Moreno, Josep Sánchez de Toledo, Soledad Gallego, Jordi Bové, Jacob M. Hooker, Miguel F. Segura

**Affiliations:** 1Group of Translational Research in Child and Adolescent Cancer, Vall d’Hebron Research Institute (VHIR)-Universitat Autònoma de Barcelona (UAB), 08035 Barcelona, Spain; roberta.antonelli@vhir.org (R.A.); carlos.jimenez@vhir.org (C.J.); aroa.soriano@vhir.org (A.S.); josep.roma@vhir.org (J.R.); lucas.moreno@vhebron.net (L.M.); jsanchezdetoledo@iconcologia.net (J.S.d.T.); sgallego@vhebron.net (S.G.); 2Martinos Center for Biomedical Imaging, Massachusetts General Hospital and Harvard Medical School, Charlestown, MA 02129, USA; mishar@nmr.mgh.harvard.edu (M.R.); JHOOKER@mgh.harvard.edu (J.M.H.); 3UOC Oncologia Pediatrica, Fondazione Policlinico Universitario “A. Gemelli” IRCCS, 00168 Rome, Italy; tiziana.servidei@guest.policlinicogemelli.it (T.S.); riccardo.riccardi@unicatt.it (R.R.); antonio.ruggiero@unicatt.it (A.R.); 4Department of Pathology, Vall d’Hebron University Hospital, Universitat Autònoma de Barcelona, 08035 Barcelona, Spain; eamartinez@vhebron.net; 5Department of Pathology, Fondazione Policlinico A. Gemelli IRCCS, Catholic University of Sacred Heart, L.go A. Gemelli, 8, 00141 Rome, Italy; maurizio.martini@unicatt.it; 6Pediatric Oncology and Hematology Department, Hospital Universitari Vall d’Hebron-Universitat Autònoma de Barcelona (UAB), Passeig Vall d’Hebron 119, 08035 Barcelona, Spain; 7Neurodegenerative Diseases Research Group, Vall d’Hebron Research Institute, Center for Networked Biomedical Research on Neurodegenerative Diseases (CIBERNED), 08035 Barcelona, Spain; jordi.bove@vhir.org

**Keywords:** pediatric brain tumors, posterior fossa ependymoma, epigenetic therapies, histone deacetylase inhibitors (HDACi)

## Abstract

Pediatric ependymoma (EPN) is a highly aggressive tumor of the central nervous system that remains incurable in 40% of cases. In children, the majority of cases develop in the posterior fossa and can be classified into two distinct molecular entities: EPN posterior fossa A (PF-EPN-A) and EPN posterior fossa B (PF-EPN-B). Patients with PF-EPN-A have poor outcome and are in demand of new therapies. In general, PF-EPN-A tumors show a balanced chromosome copy number profile and have no recurrent somatic nucleotide variants. However, these tumors present abundant epigenetic deregulations, thereby suggesting that epigenetic therapies could provide new opportunities for PF-EPN-A patients. In vitro epigenetic drug screening of 11 compounds showed that histone deacetylase inhibitors (HDACi) had the highest anti-proliferative activity in two PF-EPN-A patient-derived cell lines. Further screening of 5 new brain-penetrating HDACi showed that CN133 induced apoptosis in vitro, reduced tumor growth in vivo and significantly extended the survival of mice with orthotopically-implanted EPN tumors by modulation of the unfolded protein response, PI3K/Akt/mTOR signaling, and apoptotic pathways among others. In summary, our results provide solid preclinical evidence for the use of CN133 as a new therapeutic agent against PF-EPN-A tumors.

## 1. Introduction

Pediatric tumors of the central nervous system (CNS) are the most common solid tumors in children, second only to hematologic malignancies and are the leading cause of cancer-related deaths [1]. Despite all current therapeutic efforts, almost 50–60% of patients will not achieve a long-term cure owing to disease progression and resistance to existing therapies. This is particularly true for ependymomas (EPNs), the third most common pediatric brain tumors after astrocytomas and medulloblastomas [2].

EPNs are very heterogeneous in terms of age at diagnosis, molecular profile, histologic grade, and clinical behavior and can appear along the entire craniospinal axis, most commonly in the posterior fossa, supratentorial, and spinal cord (reviewed in [3]). Recent genomic studies based on DNA methylation profiles identified nine distinct molecular subgroups, three within each CNS compartment [4]. Among them, the most aggressive ones are *RELA* fusion-positive supratentorial EPN (ST-EPN-RELA) and posterior fossa EPN group A (PF-EPN-A) [3]. While for ST-EPN-RELA tumors, the identification of the driver oncogene and the mapping of the molecular associated alterations may facilitate the roadmap for therapeutic strategies [5,6], PF-EPN-A tumors are more heterogeneous and with low mutation rate, and no significant recurrent somatic single nucleotide variants [4,7]. However, a poor-prognosis subset of these tumors exhibits a CpG island methylate phenotype [8], which suggests that could be sensitive to epigenetic modifiers, such as polycomb repressor complex 2 (PRC2) inhibition, DNA methylation inhibitors, and histone deacetylase (HDAC) inhibitors, either alone or in combination [9].

Since a complete resection of these tumors is almost impossible and no benefits of current chemotherapies have been observed [10], in this study, we sought to screen the therapeutic potential of epigenetic drugs in PF-EPN-A models. Among them, histone deacetylase inhibitors (HDACi) were found to have the greatest anti-proliferative activity. The addition of acetyl groups from acetyl-CoA to specific lysine residues is controlled by histone acetyl-transferases (HATs) associated generally with active transcription, and removed by histone deacetylases (HDACs), which is generally connected with transcriptional repression [11].

To date, many HDAC inhibitors (HDACis) have been tested in oncology clinical trials, which led to the approval of Vorinostat, Romidepsin, Bellinostat, and Panobinostat for the treatment of cutaneous and peripheral T-cell lymphoma and multiple myeloma [12]. In brain tumors, however, tolerable doses of the HDACi Vorinostat showed no anti-tumoral effects, suggesting that it did not reach the brain at therapeutic doses [13]. Additional studies have clearly demonstrated that Vorinostat has a poor brain penetrability [14]. Owing to the large therapeutic potential of HDACis already observed in non-brain tumors, it would be desirable to find new HDACis able to reach the brain at therapeutic doses for the treatment of CNS malignancies.

Therefore, we analyzed the therapeutic potential of 5 novel brain-penetrating HDACis and found the CN133 compound to be a potent inductor of apoptotic cell death in vitro and capable of reducing tumor growth in vivo. Of note, CN133 is capable of inducing histone acetylation in orthotopically-implanted PF-EPN-A tumors at similar levels to other peripheral tissues such as the spleen showing a much better targeting of histones than other HDACis currently in clinical use. In summary, our results provide strong preclinical evidence for the use of this new HDACi, with unique brain-penetrating and safety profile, for the treatment of PF-EPN-A tumors.

## 2. Results

### 2.1. HDACi Are the Epigenetic Compounds with the Highest Therapeutic Potential in PF-EPN-A Cell Lines

Owing to the aberrant epigenetic profile of PFN-A tumors, we sought to analyze the therapeutic potential of a set of epigenetic compounds in patient-derived PF-EPN-A cell lines (i.e., EPP and EPV). Compounds capable of targeting different families of epigenetic regulators (i.e., histone-lysine methyltransferase inhibitors, bromodomain inhibitors, and histone deacetylase inhibitors) were selected for their ability to cross the blood brain barrier in animal models and with reported anti-tumoral activity in other tumors (Appendix A). For comparison purposes, we included Temozolamide (TMZ), one of the few drugs shown to be effective in preclinical EPN models [15]. Among all tested compounds, UNC1999, GSK864, TP472, and 5-aza showed anti-proliferative effects in both cell lines, albeit only at the higher tested dose (10 µM). Remarkably, the HDACi suberanilohydroxamic acid (SAHA), also known as Vorinostat, proved to be the most effective even at the lower tested concentration (Figure 1a,b).

Since the brain penetrability of Vorinostat is limited, we selected five new HDACis designed to cross the blood-brain barrier [16,17] and with different HDAC selectivity profile (Appendix A). Regardless of the specificity toward different HDACs, all compounds were able to raise the acetylation level of Histone H3 at Lys 27 (H3K27ac) in EPP and EPV cells (Figure 1c,d). Among them, CN133 and CN147 were those that impacted the most on EPP and EPV viability (Figure 1e,f).

Moreover, both inhibitors exhibited higher cytotoxicity than the FDA-approved HDACi Vorinostat (Figure 2a,b) and low toxicity levels in primary cultures of neuronal precursors (Figure 2c,d).

### 2.2. CN133 and CN147 HDACis Induce Apoptosis in EPN Cell Lines

For better understanding how CN133 and CN147 affect cell viability, flow cytometry analyses were performed. CN133 and CN147 induced a significant increase in the percentage of cells in the sub-G0/G1 peak (Figure 3a–c), one of the hallmarks of apoptotic cell death. Furthermore, an increase in the amount of the cleaved and active form of caspase-3 (i.e., 17 kD fragment) and cleavage of the caspase-3 substrate PARP (Appendix A) were also observed. Next, we analyzed another apoptotic hallmark, i.e., the exposition of phosphatidylserine in the outer membrane of apoptotic cells, a fact that can be detected with anti-annexin V antibodies. Compared to DMSO-treated cells, CN133 and CN147 increased the percentage of cells that were annexin V positive, either negative for propidium iodide (commonly referred as early apoptosis) or positive for both markers (i.e., late apoptosis). The percentage of cells only positive for propidium iodide (considered necrotic cells) did not vary significantly (Figure 3d,e). In summary, these analyses confirmed that both compounds trigger apoptotic cell death in EPN cell lines.

### 2.3. CN133 Impairs Tumour Growth and Extends Survival of Mice Bearing EPN Orthotopic Xenografts

To understand how CN133 and CN147 modulate histone acetylation in EPN cell lines and predict potential behavior in vivo, both compounds were administered in a “constant” compared with a “pulse” setting (Appendix A). Under both conditions, HDACi treatments induced a rapid increase in the acetylation of different histone residues, such as H2BK5, H3-N-term, and H3K9Me2. The hyperacetylation at all marks reached maximum levels at 4 h and remained constant as long as the inhibitor was present in the cell media. However, when EPP cell lines were treated for 6 h, washed and further incubated for up to 18–24 h in inhibitor-free media, the histone acetylation levels returned to basal levels. These results suggest that the compounds hit the targets in a transient and reversible manner. Thus, repetitive treatments would be needed to maintain high histone acetylation levels in vivo.

Next, to identify possible adverse effects resulting from exposure to these compounds in vivo, six-week-old NMRI nude mice were treated i.p. daily with either 1 mg/kg or 5 mg/kg for 1 week either once or twice a day. Average mouse weight showed no major differences between the control and compound-treated groups with either one or two injections per day (Figure 4a,b).

To determine the anti-tumoral efficacy of CN133 and CN147 in vivo, EPN orthotopic xenografts were established by stereotaxic implantation of luciferase-expressing EPN cells into the fourth ventricle of nude mice. Cells were left to engraft and form a tumor for a period of 10 days (Appendix A) and then, mice were randomized into three groups, namely, vehicle (*n* = 13), CN133 (*n* = 13), and CN147 (*n* = 13), and treated twice a day for 40 days. Tumor growth was monitored by in vivo bioluminescent imaging once a week. While tumors of vehicle-treated mice grew very fast, those tumors of HDACi-treated grew at a much slower pace (Figure 4c–f). Statistically significant differences were appreciated starting day 21 of treatment, with CN133 being more effective than CN147 (Figure 4g). However, only CN133 significantly prolonged the animal’s lifespan (*p* = 0.0019, Figure 4h,i). Collectively, these data suggest that CN133 exerts a better therapeutic effect than CN147 on EPN in vivo.

Finally, to confirm that CN133 was able to cross the blood–brain barrier and target HDACs in the implanted tumor cells more effectively than the other HDACis, mice were treated for 14 days with either 5 mg/kg CN133 or 50 mg/kg SAHA (Vorinostat), the most commonly-used HDACi for the treatment of brain tumors. A peripheral tissue (i.e., spleen) and brain tumors were collected 4 h after the last drug administration and histone acetylation was analyzed. H2B resulted dramatically hyperacetylated at residue H2BK5Ac in both spleen and brain tumors of CN133-treated mice, while SAHA treatment only mildly affected H2B acetylation levels (Figure 5a–d). Of note, H2B hyperacetylation levels induced by CN133 were very similar in EPN tumors and spleen, which further supports the good brain penetrability profile of CN133. These findings corroborate the evidence that CN133 is a much more potent HDACi than Vorinostat in increasing histone acetylation in brain tumors, and places CN133 as a new promising therapeutic candidate for ependymoma treatment.

### 2.4. CN133 Modulates the Expression of Unfolded Protein Response, PI3K/AKT/mTOR, and Apoptosis-Related Genes in EPN Cell Lines

HDACs are known to play crucial roles in cancer by de-acetylating histones and non-histone substrates, leading to altered expression of genes involved in several cellular processes, such as cell cycle control, apoptosis, DNA-damage response, angiogenesis, and autophagy, among others [18]. To shed light on the transcriptomic changes induced by CN133 treatment on EPN cells, a microarray expression analysis of the EPP cell line treated with CN133 was performed. Principal component analysis (PCA) showed remarkable segregation of distinct expression profiles between vehicle (DMSO) and CN133-treated cells (Appendix A). A total of 2429 genes were found to be differential expressed with a fold change <−2 or >2 and a false discovery rate <0.05 (1145 upregulated and 1284 downregulated). Some differentially-expressed genes were validated either by quantitative real-time PCR (Figure 6a) or by western blot (Figure 6b). Gene set enrichment analysis (GSEA) revealed that differentially-expressed genes were associated with the unfolded protein response (UPR), hypoxia, PI3K/Akt/mTOR signaling, and apoptosis (Figure 6c and Appendix A). Key components of these pathways were analyzed by western blot. We observed that the hypoxia marker HIF-α, together with several components of the UPR pathway, such as the pancreatic eIF2-α kinase (PERK), activating transcription factor 3 (ATF-3) and ATF-4, were upregulated in response to CN133 treatment. In addition, cleavage of ATF-6, upregulation of transcription factor C/EBP homologous protein (CHOP) and the cell cycle inhibitor p21 were also observed (Figure 6d), thereby confirming activation of UPR.

In relation to the PI3K/AKT/mTOR pathway, a reduction in levels of phosphoAKT_S473_, phospho-GSK3β and phospho-S6 levels suggested a clear inhibition of the pathway (Figure 6e). Finally, key proteins of the apoptotic program were also analyzed. While pro-apoptotic protein Bim and the anti-apoptotic protein Mcl-1 were not altered in the presence of CN133, a clear upregulation of the pro-apoptotic BH3-only Bcl-2 family member Noxa was observed, with concomitant downregulation of the anti-apoptotic protein X-linked inhibitors of apoptosis (XIAP). Increases in the cleaved form of caspase-3 and its substrate PARP were evident after 24 h of CN133 treatment (Figure 6f) thereby confirming the activation of the apoptotic program.

In summary, this evidence suggests that CN133 increases in the histone acetylation levels that consequentially modulate the unfolded protein response (UPR), hypoxia, PI3K/Akt/mTOR signaling, and apoptosis pathways to slow cell cycle progression and/or trigger apoptotic cell death programs (Figure 6g).

## 3. Discussion

Recent genomic and transcriptional analyses showed that the most aggressive and frequent subtypes of EPN, such as PF-EPN-A, harbor very few recurrent genetic alterations while possessing a significant proportion of frequent events converging on epigenetic mechanisms [19]. These findings have may open up new avenues for PF-EPN-A therapy by targeting epigenetic modifications.

Several clinical trials on pediatric patients showed epigenetic drugs to be safe and well tolerated [20,21,22]. The present study sought to analyses the therapeutic potential of preclinical-validated epigenetic drugs known to exhibit antitumor activity in brain tumor models and identified targets (i.e., histone-lysine methyltransferase inhibitors, bromodomain inhibitors, and HDACis). This screening performed in patient-derived PF-EPN-A cell lines showed HDACis to be the epigenetic compounds with the highest therapeutic potential.

In recent years, altered expression and mutations of genes encoding for HDACs have been linked to tumor development since, together with histone acetyltransferases, they concur to promote an aberrant transcription of key genes involved in controlling cell proliferation, cell-cycle regulation and apoptosis. Mining mRNA microarray expression data for ependymoma (GSE64415, *n* = 209) showed that *HDAC1*, *HDAC2*, and *HDAC3* mRNA expression levels were significantly upregulated in EPN compared to CNS tissues (i.e., whole brain GSE11882, *n* = 172; cerebellum GSE3526, *n* = 9), thus underlining a potential role for these enzymes in the development of this particular type of pediatric brain tumor (Appendix A). The anti-tumor efficacy of pan-specific HDACis was investigated and promising results were obtained by treating patients with some FDA-approved drugs, such as Vorinostat, Romidepsin, Bellinostat, and Panobinostat. In particular, Vorinostat has been approved for the treatment of cutaneous T-cell lymphoma, and Panobinostat for multiple myeloma. Despite promising preclinical evidence in EPN models [23], HDACis did not yield the expected clinical results in patients with CNS tumors, most probably due to their inability to reach and accumulate in the brain at the concentration required to elicit a therapeutic response [18].

Here, five new HDACis proved to impact on PF-EPN-A cell lines viability even more robustly than the FDA-approved Vorinostat. All compounds were able to strongly enhance the acetylation levels of Histone H3, CN133 and CN147 being the most effective. Interestingly, pharmacokinetics and brain distribution analyses revealed that both compounds are characterized by an excellent brain accumulation profile, reaching brain-to-plasma ration of 20:1 upon intraperitoneal administration. Furthermore, they demonstrated strong potency and isozyme selectivity for HDAC1, HDAC2, HDAC3, HDAC4, HDAC5, and HDAC6 in a recombinant human enzyme assay (Appendix A).

When these compounds were used to treat orthotopically-implanted PF-EPN-A xenografts, both CN133 and CN147 significantly delayed tumor growth, although only CN133 was able to prolong the lifespan of treated animals. We believe these differences were not linked to the ability of the compounds to reach and accumulate in the brain, since both showed the same brain-to-plasma partition of 20:1, but to a difference in potency to inhibit HDACs. In fact, CN133 exhibited an IC_50_ for HDAC1, 2, and 3, which is almost 500 times lower as compared to that of CN147. Moreover, while CN133 reduced EPN cell viability in dose-dependent manner, CN147 showed cytotoxic activity when overcoming a specific concentration threshold, kinetic behavior quite difficult to control in the in vivo scenario.

Understanding how HDACis exert their therapeutic effect is complex. First, HDACs associates with distinct multi-subunit complexes which exhibit diverse, and often, cell type-specific functions. These HDAC interacting partners not only modulate the catalytic activity and substrate specificity of the histone deacetylase(s) present in the multiprotein complex, but also alter the accessibility of catalytic sites of HDACs for the inhibitor itself. Second, HDACs can also interact with each other, adding further complexity to interaction network. Third, although the name HDAC implies some specificity for histones, HDACs are known to affect the acetylation status of a wide variety of non-histone proteins, including transcription factors and chaperones, among others, resulting in changes in protein stability, protein-protein, and protein-DNA interactions [24].

Transcriptome and gene set enrichment analyses showed that the CN133 compound modulates several cell-signaling pathways in EPN cells which ultimately lead to cell cycle arrest and apoptosis. In particular, the cyclin-dependent kinase inhibitor p21 (*CDKN1A*) appears to be one of the most robustly and rapidly up-regulated genes. p21 overexpression may lead to cell cycle arrest in G1 phase and induction of apoptosis (reviewed in [25]), a mechanism shown to occur in other brain tumor cell lines treated with different HDACis, such as Vorinostat, Trichostatin A [26], Entinostat [27], Dacinostat, or Sodium butyrate [28]. Although the activation of p53 usually may be responsible for the upregulation of p21 in response to drug treatments, different mechanisms have been proposed to be responsible for the upregulation of p21 upon HDACi treatment. In particular, Richon et al. reported that HDACis, such as SAHA, were able to increase the acetylation of histones at specific genomic sites, such as the p21 locus, thereby inducing the direct transcription of p21 [29] in a p53 independent manner. Our evidence supports this mechanism, since CN133 is able to increase p21 even when p53 is silenced (Appendix A), or in other cell lines with non-functional p53 (Appendix A). CN133 also negatively impacts on the PI3K/AKT/mTOR pathway, as evidenced by the reduction in phosphorylated levels of AKT_S473_ and the downstream target ribosomal protein S6. Similar results have been observed in B-lymphoma cells treated with the HDACi MPT0E028 [30].

Moreover, CN133 also modulates the expression of several factors of the unfolded protein response (UPR) pathway. The CN133-dependent induction of UPR could be mediated by modulation of hypoxia-related genes (i.e., HIF-1α) and/or by the direct acetylation of glucose-regulated protein 78 (GRP78), a chaperone molecule assisting the folding, assembly, and translocation of newly-synthesized polypeptides across the ER membrane. During ER stress, the dramatic increase in unfolded substrates leads to the sequestration of GRP78, releasing sensors, such as PERK to initiate UPR signals [31,32]. ER-stress activation mechanism leading to apoptotic cell death has also been observed in neuroblastoma cells treated with SAHA [31]. UPR-related proteins, such as ATF3, ATF4, and CHOP were upregulated at 24 h post-treatment, a phenomenon that could be sufficient to trigger apoptotic cell death. In fact, downregulation of anti-apoptotic protein XIAP at earlier time points and the concomitant upregulation of the pro-apoptotic protein NOXA, would confirm orchestrated induction of the apoptotic cascade. A similar chain of events has been described for glioma or neuroblastoma cells treated with HDACi [33,34].

Although an optimization of drug dosing and administration schedules could significantly improve the therapeutic potential of CN133 as a monotherapy, a combination of epigenetic compounds with standard drugs or radiation regimens is often proposed in clinical trials [35]. For example, the combination of the HDACi Vorinostat and Valproic acid with TMZ and/or radiotherapy in CNS malignancies such as diffuse intrinsic pontine glioma (DIPG) or adult GBM improved the outcomes in the patients with a 1-year OS rate of 86% (CI: 76–98) and a 6-month PFS rate of 70% (CI: 57–87) [36]. Thus, the use of CN133 in combinations with standard therapies could improve the patient’s outcome, a fact that warrants further investigation.

## 4. Materials and Methods

### 4.1. PF-EPN-A Cell Lines

The EPN cell lines, EPP and EPV, were derived from the recurrence of two pediatric infratentorial ependymoma [37] which showed the typical EPN histological pattern (Appendix A). EPP was derived from a 34 month-old male diagnosed with a grade II infratentorial EPN; EPV was derived from a 23 months-old male diagnosed with a grade III infratentorial EPN. Both samples were retrieved in accordance with the institutional Catholic University of Rome board approval (ID1648; Prot. 44510/17). To distinguish between PF-EPN-A and PF-EPN-B, H3K27me3 immunohistochemistry [38,39] was performed and evaluated independently by two observers. Immunostainings of both patient samples were shown to be negative for H3K27me3, thereby suggesting that both tumors were PF-EPN-A (Appendix A). Furthermore, both tumors showed high expression levels of Laminin alpha 2 (LAMA-2) and Tenascin (TNC), two gene expression markers shown to be upregulated in PF-EPN-A vs. PF-EPN-B (Appendix A, [7]). Moreover, low expression levels of neural epidermal growth factor like-2 (NELL2) and Kinesin Family Member 27 (KIF27) confirmed that EPP and EPV patient cell lines origin from PF-EPN-A tumors.

EPP and EPV patient-derived cells were always cultured in 3D conditions using Neurocult medium supplemented with NeuroCult NS-A Proliferation Supplement Human (Stem Cell Technologies, Vancouver, BC, Canada), Epidermal Growth Factor (EGF; 20 ng/mL; Stem Cell Technologies), and basic Fibroblast Growth Factor (bFGF; 10 ng/mL; Stem Cell Technologies) and maintained at 37 °C and in a 5% CO_2_ saturated atmosphere. DNA-methylation profiled was performed, but as expected for the majority of cultured EPN cell lines [40], results were not conclusive.

### 4.2. Chemical Compounds

GSK2801, JQ1, UNC0642, UNC1999, BAY 598, GSK864, GSK484 and Tubastatin were provided by Sigma-Aldrich (St. Louis, MO, USA); TP-472 (Structural Genomics Consortium); 5-Azacytidine and SAHA from Abcam (Cambridge, UK); CN26, CN101, CN133, CN147, and C161 (Hooker laboratory). Unless otherwise indicated, all compounds were dissolved in DMSO.

### 4.3. Cell Viability Assay

EPP and EPV were dissociated and 3.6 × 10^4^ cells per well were seeded in 12-well plates. Twenty-four hours later, cells were treated with 500 nM, 1 μM and 10 μM of selected drugs (Appendix A) or dimethyl sulfoxide (DMSO) as control. Seventy-two hours after treatment, spheres were dissociated and Alamar Blue Cell Viability Reagent (Thermo Fisher Scientific, Whaltham, MA, USA) added. Six hours after incubation, samples were read at 544 and 590 nm (excitation and emission wavelengths, respectively) using an Appliskan Multimode Microplate Reader (Biotek, Winooski, VT, USA).

### 4.4. Western Blot

Cells were harvested in RIPA buffer (Thermo Fisher Scientific, Whaltham, MA, USA) supplemented with EDTA-free complete protease inhibitor cocktail (Roche, Basel, Switzerland), phosphatase inhibitors sodium fluoride, and sodium orthovanadate. Cell lysates were resolved on 4–12% tris-glycine SDS polyacrylamide electrophoresis gels (Thermo Fisher Scientific) and transferred onto nitrocellulose membranes. Membranes were blocked for 1 h with 5% non-fat milk or 5% bovine serum albumin and probed with the indicated antibodies Appendix A. Band intensity was quantified with ImageJ software 1.x (Bethesda, Maryland, USA).

### 4.5. Cell Death Assay

One million EPP and EPV cells were incubated with 1 µM CN133 and CN147 at indicated times. Pellets were resuspended in staining solution (38 mM Sodium citrate (Sigma-Aldrich), 500 μg/mL Propidium iodide (Thermo Fisher Scientific), 10 mg/mL RNAse A (AppliChem, Darmstadt, Germany) in PBS and incubated overnight at 4 °C. Stained cells were subjected to flow cytometric analysis on FACScan (BD Biosciences, San José, CA, USA) and cells in Sub G1 peak were quantified. Data were analyzed with FACS Diva 6.0 (BD Biosciences) and CellQuest Pro software (BD Biosciences).

### 4.6. Annexin-V FITC/PI Assay

One million EPP and EPV cells were incubated with 1 µM CN133 and CN147 for 24 h. At the end of the treatment, cells were harvested and stained with the Annexin V-FITC Apoptosis Staining Detection Kit (abcam ab14085) following the manufacturer’s instructions. The apoptosis events were analyzed by flow cytometry (BD LSRFortessa, BD Biosciences) using untreated cells as negative control for gating. 10,000 events were recorded. The percentage of live, early apoptotic, late apoptotic, and necrotic cells were analyzed using FCS Express v 4. (De Novo software, Pasadena, CA, USA).

### 4.7. Ependymoma Orthotopic Mouse Xenografts

All animal procedures were approved by the ethical committee of Vall d’Hebron Research Institute (protocol number 98.17). To generate orthotopic xenografts, 3 × 10^5^ firefly luciferase-transduced EPP cells in 10 µL PBS were implanted into the fourth ventricle of NMRI-nude mice using stereotaxic injection (Kopf Instrument, Tujunga, CA, USA). The injection coordinates were: anteroposterior, 6.0 mm from Bregma; lateral to right, 0.2 mm; dorsoventral, 4 mm. For drug efficacy studies, mice were randomized ten days post-injection into drug-treated and control groups (*n* = 13 mice/group), then treated i.p. with 5 mg/kg CN133, 5 mg/kg CN147, or vehicle (10% DMSO, 10% Tween-20, 80% NaCl 0.9%) twice a day for 46 days. Tumor growth was monitored by bioluminescent imaging. Images were captured using IVIS imaging system (Spectrum CT; PerkinElmer, Waltham, MA) and quantified using Living Image Software 4.5.2(Perkin Elmer). Total Flux (Photons/second) was normalized to background signal.

For in vivo histone acetylation studies, six-week-old NMRI nude mice were treated with 5 mg/kg of CN133, 50 mg/kg SAHA or vehicle by i.p. twice a day for 14 days (*n* = 4 mice/group). At the end of the treatment, the spleen and brain were collected. Tumor cells were isolated using a scope equipped with epifluorescence. Tissues were processed using the Histone Extraction Kit (ab113476). Histone-enriched lysate were resolved on 4–12% tris-glycine sodium dodecyl sulphate polyacrylamide electrophoresis gels (Thermo Fisher Scientific). H2BK5Ac was used as a histone acetylation marker.

### 4.8. Transcriptome Analyses

EPP cells (3 × 10^5^ cells per well) were treated with 1 µM of C133 or DMSO for 5 h. RNA was isolated using the miRNeasy mini kit (Qiagen, Germantown, MA, USA). RNA quality was evaluated using Agilent Bioanalyzer 2100 Eukaryote Total RNA Pico assay (Agilent Technologies, Santa Clara, CA, USA). A total of 300 nanograms of total RNA were hybridized to Human Clariom^TM^ S assay (Affymetrix, Santa Clara, CA, USA) arrays with the GeneChip WT Terminal Labelling and Hybridization Kit (Affymetrix). The expression analysis was performed using the transcription analysis console version 4.0.0.25 (Thermo Fisher Scientific) and functional annotations of resulting gene lists were performed using GSEA software [41]. The accession number for microarray analysis reported in this paper is GSE139210.

### 4.9. Real-Time Quantitative PCR

Total RNA was extracted from cell lysates using RNeasy Mini Kit (Qiagen). A total of 1 μg of RNA was subjected to DNase treatment and retrotranscription using the High-Capacity cDNA reverse transcription kit (Thermo Fisher Scientific). Real-time PCR of *SERPINE1*, *HMOX1*, *CXCR4*, *EFNA3*, *GADD45b*, *GSKIP*, *EGR1 HAS1*, and *LRRC4b* genes was performed using SYBR green fluorescence (Thermo Fisher Scientific). L27 was used as internal standard. Primers are listed in Appendix A. Relative quantification of gene expression was performed with the 2^-ΔΔCt^ method [42].

### 4.10. siRNA Transfection

5 × 10^5^ EPP cells were transfected with 25 nM of p53 siRNA (J-003329-16 and J-003329-17 Dharmacon, Horizon discovery, Lafayette, CO, USA) or 25 nM of BLOCK-iT™ Fluorescent Oligo (Thermo Fisher Scientific) using Lipofectamine RNAiMAX (Thermo Fisher Scientific).

### 4.11. Statistical Methods

Unless otherwise stated, graph values are the mean ± SEM of three independent experiments. Statistical significance was determined by unpaired two-tailed Student’s t test (GraphPad Prism 6 Software, La Jolla, CA, USA).

## 5. Conclusions

Pediatric EPN, and in particular those classified as PF-EPN-A, are very aggressive tumors in demand of new therapeutic approaches. Cumulative evidence supports the use of epigenetic therapies for the treatment of cancer either as single agents or in combination with standard therapies. However, patients with brain tumors have not yet benefit from these therapies, most probably due the limited drug biodistribution. Here, we provide a strong preclinical evidence for the use of CN133 as a new HDACi for the treatment of pediatric EPN owing to this cytotoxic potential compared to non-tumoral cells, high brain penetrance, and low toxicity.

## Figures and Tables

**Figure 1 cancers-12-01922-f001:**
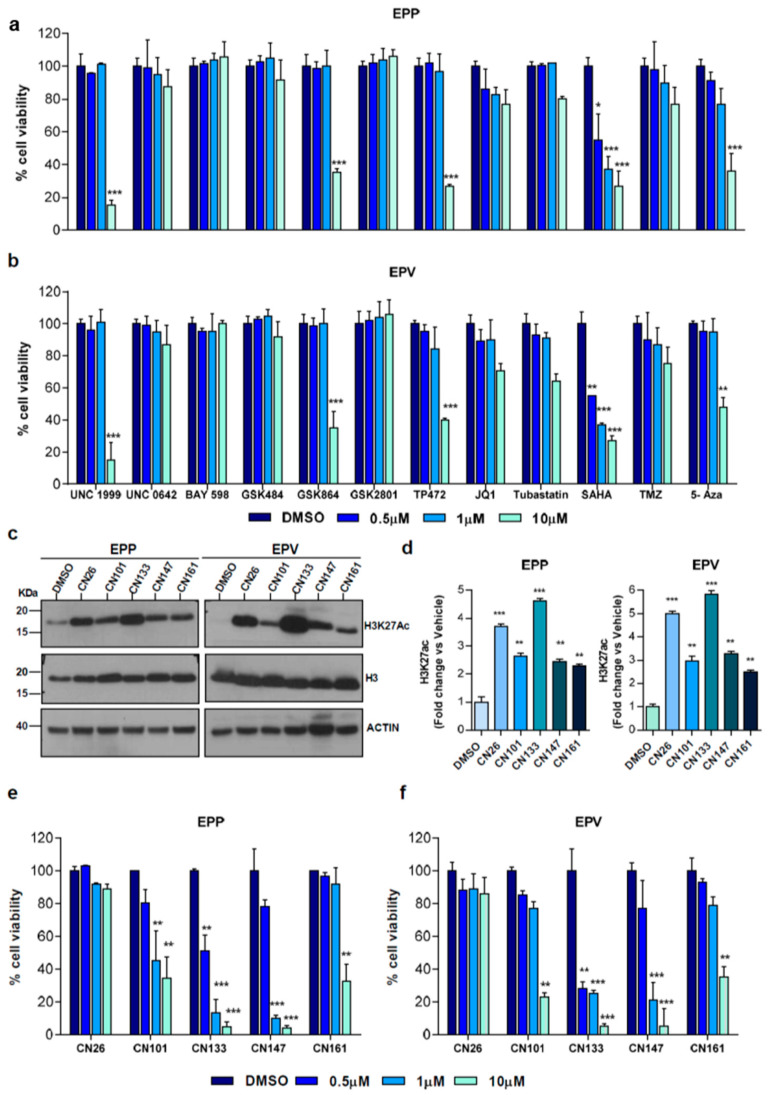
Epigenetic drug screening in patient-derived EPN cell lines: (**a**) EPP or EPV (**b**) cell lines were treated with either vehicle (DMSO) or the indicated drugs at 0.5 μM, 1 μM or 10 μM for 72 h. Cell viability was evaluated using Alamar Blue; (**c**) Western blot of histone acetylation levels upon treatment with the indicated compounds (1 μM) for 48 h in EPP and EPV cell lines. Uncropped Blots of Figure 1c are shown in Appendix A (**d**) graph represents the quantification of (**c**); H3K27ac were normalized versus total H3 and expressed as a fold change versus DMSO-treated cells; five experimental brain-penetrating HDACis were tested in cell viability assays on EPP (**e**) and EPV (**f**). Each graph represents the average of three independent experiments (*n* = 6/each experiment) ± SEM ** *p* < 0.01; *** *p* < 0.001.

**Figure 2 cancers-12-01922-f002:**
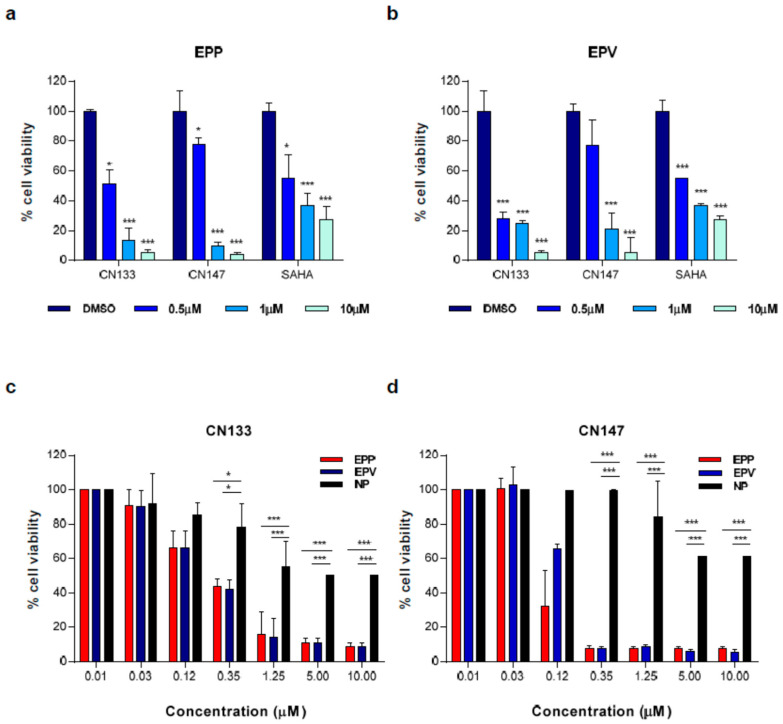
CN133 and CN147 have stronger anti-tumoral activity than SAHA and are more cytotoxic in tumor cells than in non-transformed cells. Cell viability assay in EPP (**a**) or EPV (**b**) cell lines with either vehicle (DMSO) or with the indicated HDACi at 500 nM, 1 μM or 10 μM for 72 h. Cell viability was evaluated using Alamar Blue reagent. Graph represents the average of three independent experiments. (**c**,**d**) Cell viability assays in EPP, EPV or neuronal progenitors (NP) treated with vehicle (DMSO) or the indicated doses of CN133 (**c**) or CN147 (**d**) for 72 h. Cell viability was evaluated using Alamar Blue reagent. Each graph represents the average of three independent experiments (*n* = 6/each experiment) ± SEM. * *p* < 0.05; ** *p* < 0.01; *** *p* < 0.001.

**Figure 3 cancers-12-01922-f003:**
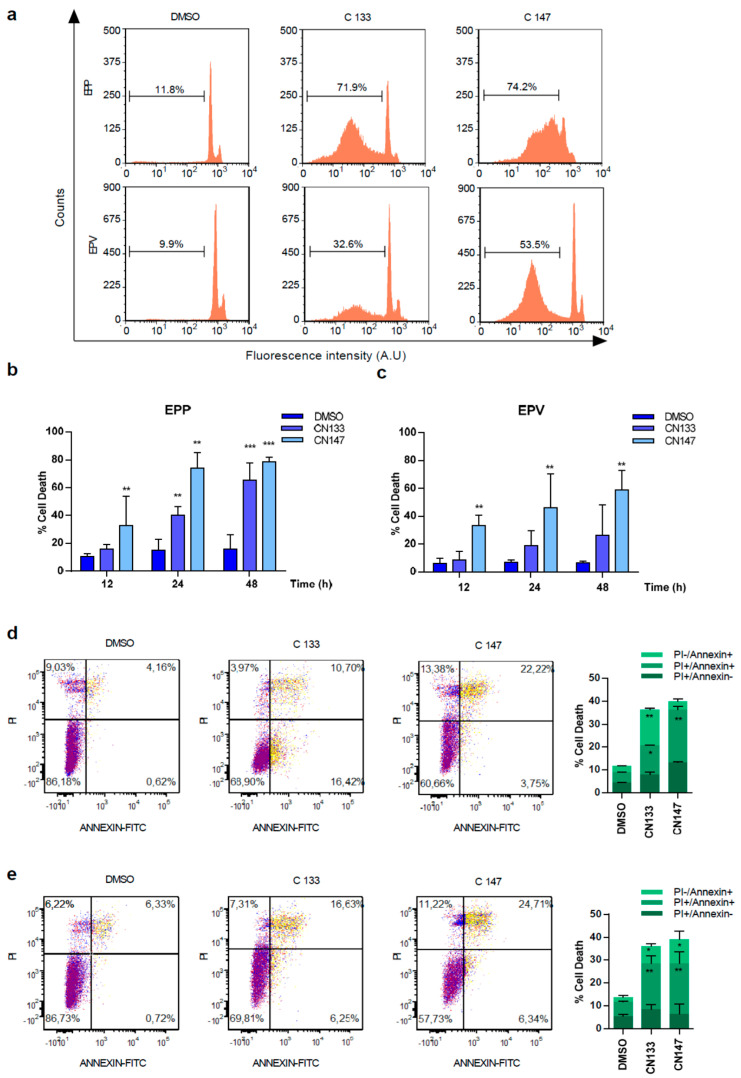
CN133 and CN147 induce cell death in EPN cell lines: (**a**) EPP and EPV cell lines were treated with 1 μM of CN133 and CN147 and apoptosis was evaluated by flow cytometry (FACS); (**b**,**c**) Graph represents the average quantification of the percentage of cells in sub-G1 of three independent experiments (*n* = 3/experiment) ± SEM. ** *p* < 0.01; *** *p* < 0.001; EPP (**d**) and EPV (**e**) were treated with 1 µM of CN133 and CN147 for 24 h. At the end of the treatment, cells were harvested and stained with the Annexin V kit following the manufacturer’s instruction. Graph represent the average quantification (*n* = 3/each). * *p* < 0.05; ** *p* < 0.01; *** *p* < 0.001.

**Figure 4 cancers-12-01922-f004:**
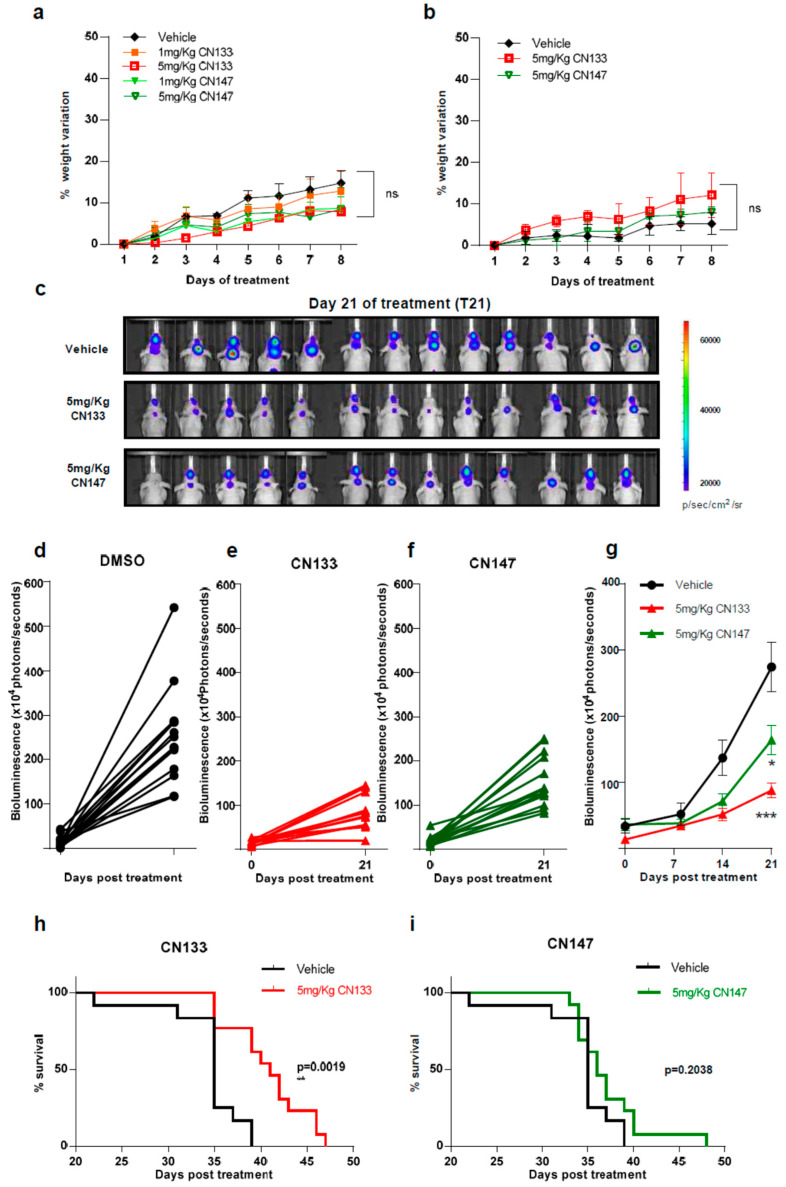
CN133 delays EPN tumors growth and extends survival of mice bearing EPN orthotopic xenografts. Percentage of weight variation of nude mice treated with 1 or 5 mg/kg of CN133 or CN147 for 8 days by i.p. administration (*n* = 5 mice/group) once (**a**) or twice a day (**b**); (**c**) representative bioluminescent images at the indicated days of EPP luciferase-transduced cells orthotopically-injected in the posterior fossa of nude mice; (**d–f**) bioluminescence quantification of EPP tumors progression at 21 days post treatment DMSO (**d**) C133 (**e**) and C147 (**f**); (**g**) bioluminescence quantification of EPP tumors over time (**h**,**i**) Kaplan–Meier survival curves of mice implanted with EPP luciferase-transduced cells (*n* = 13/group) and treated with vehicle or the indicated drugs.

**Figure 5 cancers-12-01922-f005:**
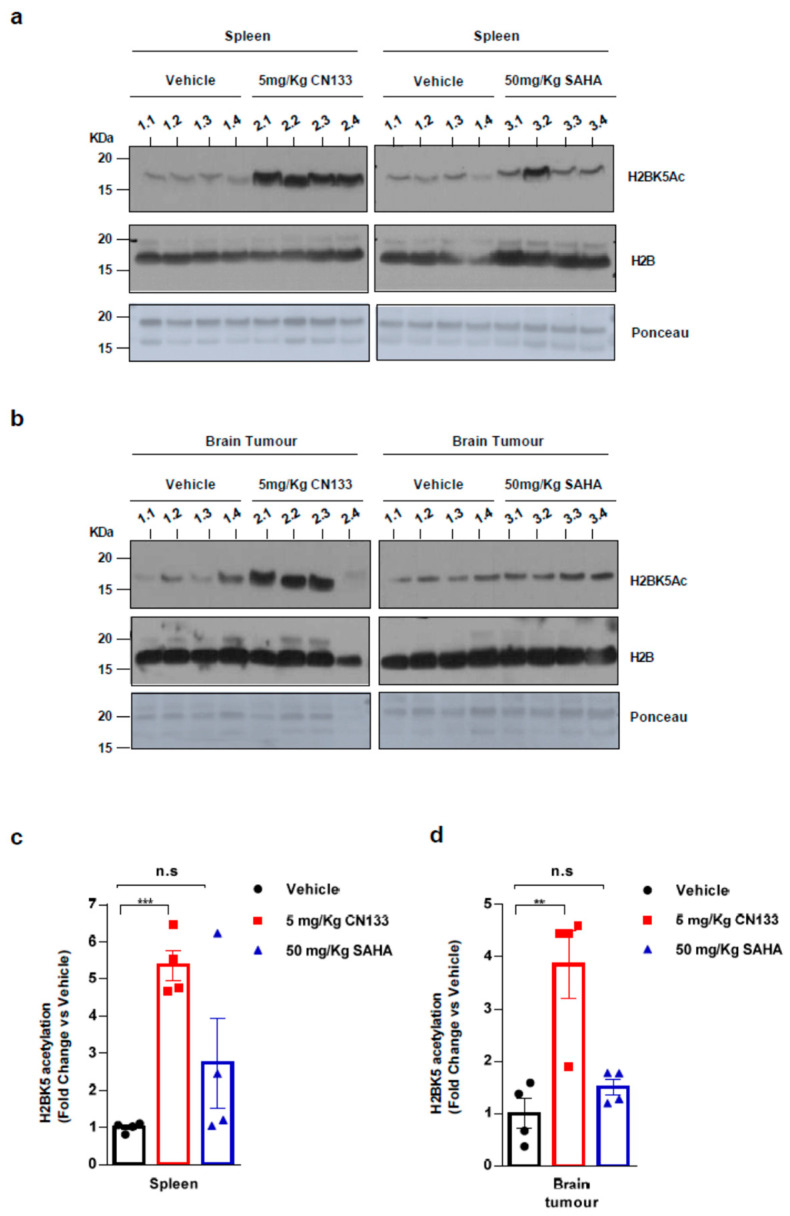
CN133 is more potent than SAHA in increasing histone acetylation in vivo: Western Blot of spleen (**a**) or brain tumors (**b**) from mice treated for 14 days with 5 mg/kg C133 or 50 mg/kg SAHA administered twice a day by i.p. Numbers above the western blots indicate mouse identification. Uncropped Blots of Figure 5a,b are shown in Appendix A (**c**,**d**) Densitometry analysis of H2BK5ac levels normalized to H3. Fold changes are relative to values from vehicle-injected mice. ** *p* < 0.01; *** *p* < 0.001.

**Figure 6 cancers-12-01922-f006:**
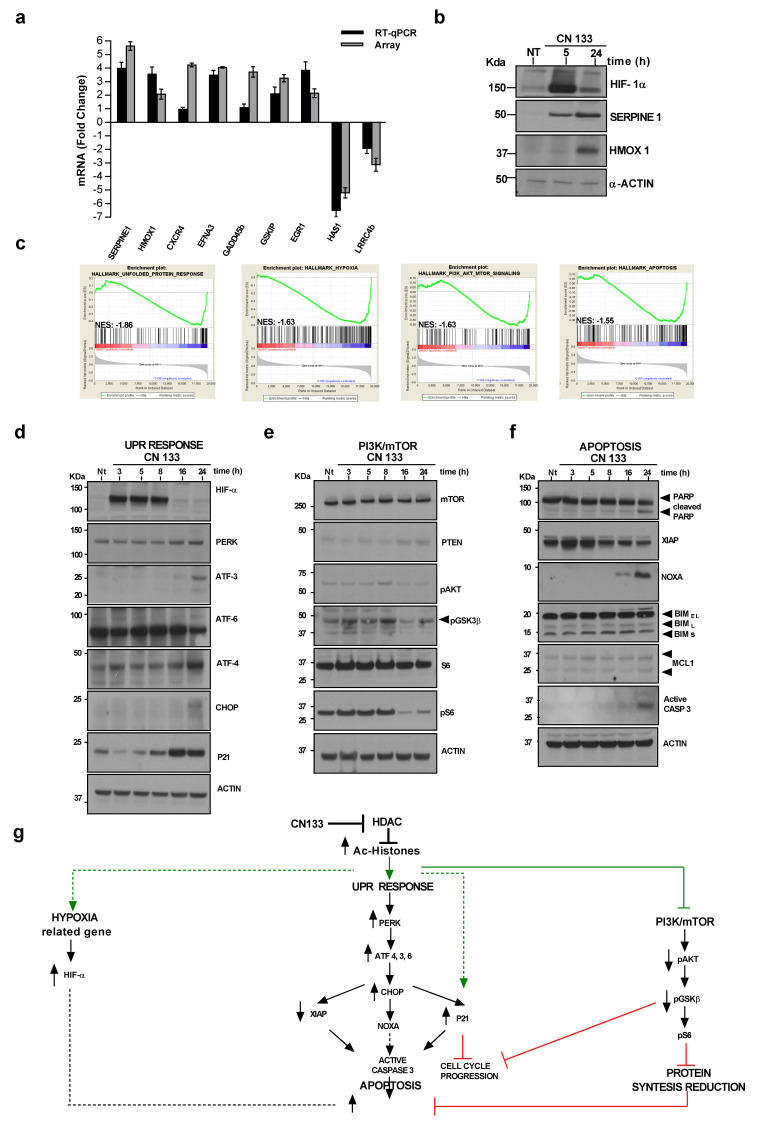
CN133 modulates several cancer-related pathways. Uncropped Blots are shown in Appendix A (**a**) Quantitative real-time PCR (RT-qPCR) using primers for seven upregulated genes and two downregulated genes. Values are represented as fold change of treated versus non treated cells and are the mean ± SEM of three independent experiments; (**b**) Western blot of some upregulated genes in response to CN133; (**c**) gene set enrichment analysis GSEA performed on genes differentially expressed in presence or absence of CN133 treatment on EPP cell line; (**d**) Western blot of the proteins related to hypoxia (HIF1α) or to the unfolded protein response from EPP cells treated with 1 μM of CN133 for the indicated times. Actin was used as a loading control; (**e**) Western blot of proteins related to the mTOR/AKT pathway; (**f**) Western blot of apoptotic-related proteins; (**g**) schematic representation of the proposed mechanism of action of CN133.

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
