# Peer review of "CN133, a Novel Brain-Penetrating Histone Deacetylase Inhibitor, Hampers Tumor Growth in Patient-Derived Pediatric Posterior Fossa Ependymoma Models"

_cancers, 2020, doi:10.3390/cancers12071922_

Round 1

Reviewer 1 Report

The manuscript has been improved.

However, based on my first report, flow cytometry analysis based on PI-annexin should be performed to quantify apoptosis at the cellular level. 

The analysis by PI-quantification alone has many drawbacks. The main disadvantage can be seen in the histograms in Figure3a of the revised manuscript, in which the subG1 percentage of untreated (or DMSO-treated) cells is >10%, which is high enough for healthy cells. Normally, in a cultured cell population subG1 cells are up to ~5-7%. A higher percentage may be an artefact of the method, or that the intrinsic cell death rate of these cells are higher than most cell cultures. This means that the more accurate method of PI-annexin is necessary to accurately quantify this phenomenon.

I want to stress out that the molecular and biochemical results of this study support the induction of apoptosis. The authors have invested their time and effort to clearly prove cell death induction. For such a high quality study design and in order these results to be confirmed at the cellular level , I insist that apoptosis should be quantified by PI-annexin.

Author Response

We thank the reviewer #1 for his/her comment that "The manuscript has been improved".

We agree with the statment "...subG1 percentage of untreated (or DMSO-treated) cells is >10%, which is high enough for healthy cells. Normally, in a cultured cell population subG1 cells are up to ~5-7%. A higher percentage may be an artefact of the method, or that the intrinsic cell death rate of these cells are higher than most cell cultures". and following the reviewer's suggestion we performed Annexin V/PI staining in our cells treated with CN133 and CN147 compounds.

Concurring with our previous observations, there is a basal cell death (both, necrotic and apoptotic in our cell lines) a little bit above 10%. These cell cultures grow in suspension and forming spheroid aggregates. In order to get single cells for FACS analysis, these spheres have to be enzymatically (i.e. accutase treatment) and mechanically dissociated, a phenomenon that clearly can affect the integrity of the cells, compared with other type of adherent or cell suspension cultures.

Despite this levels of basal cell deaht, the Annexin V results confirmed our previous observations, showing a clear increase in the % of apoptotic cells upon CN133 and CN147 treatment. In the case of EPP cells, both compounds also showed a minor increase in the fraction of necrotic ells, but these differences were not statistically significant.

These results have now been included in Figure 3 (panels d-e), in the results section (lines 170-177) and the associated methodology (lines 444-450).

We thank once more the reviewer for his/her suggestion and we hope that our manuscript is now suitable for publication.

Reviewer 2 Report

All my concerns have been adequately addressed.

Author Response

We thank the reviewer for all his/her suggestions that have improved our manuscript

Reviewer 3 Report

The revised manuscript is suitable for publication now.

Author Response

We thank the reviewer for all his/her suggestions that have improved our manuscript

This manuscript is a resubmission of an earlier submission. The following is a list of the peer review reports and author responses from that submission.

Round 1

Reviewer 1 Report

The authors tested in vitro and in vivo the effects of 5 new brain penetrating hystone deacetylase inhibitors on two ependimoma derived primary cell cultures. They convincingly demonstrated both in vitro and in vivo after xenotransplantation, that one of the screened compunds (C-133) had good inhibitory activity against human ependymoma cells.
Since their goal was testing new HDACi able to penetrate the blood brain barrier (BBB) I find bizare that they compared the effect of the new compounds against vorinostat and did not enclosed in the comparison valproic acid, an inhibitor that certainly pass the BBB. Inclusion of data on the effect of valproic acid in their experimental setting will certainly improve the paper.
Were the detailed results of the transcriptional analysis uploaded to a public repository? If not, please explain why not?

Reviewer 2 Report

The manuscript “CN133, a novel brain-penetrating histone deacetylase inhibitor, hampers tumor growth in patient-derived pediatric posterior fossa ependymoma”, by Antonelli et al. This study reveals the action of an HDAC inhibitor, selected among several molecules with simiral action, that show significant anticancer activity. The results of the study look convincing and I would like to congratulate the authors on their step-by-step approach. The manuscript is easy to follow in most sections. I have comments on two sections that I believe need some impromevent.

  1. The rationale of the study needs further explanation in the first section of the results. The study starts with several molecules and the ones that have the most significant action belong to the family of HDAC inhibitors. How exactly compounds in Figure 1a and b were selected? The authors should explain in more detail the connection between the compounds in Figure 1 panels a and b and the CN-XXX compounds mentioned in figure 1c-f. If I understand correctly, molecules in panels a and b don’t pass the BBB, that’s why CN-xxx were selected, but this should be clarified in the text.
  2. Although, sub-G1 peak quantification after PI-staining may support induction of cell death, a more reliable method is that of PI-annexin staining. In that way the authors will achieve the quantification of early and late apoptotic cells and also the possible induction of necrotic death. This method should be used in the assay conditions of Figure 3.

Reviewer 3 Report

The authors have submitted an interesting manuscript about a brain-targeting HDAC inhibitor. The provided experiments are accurate and the good results warrant publication. I have the following suggestions for a revised version:

Line 87: There are two full stops at the end of the sentence.

Figures 1, 5, and 6: Letters and numbers are distorted. Please improve the readability of the figures in the revised version.

Cell cycle inhibitor p21 was strongly upregulated by CN133. Please discuss the provided cell cycle analyses of cells treated with CN133 (cell accumulation in sub-G0/G1) in the light of this finding. Can the authors provide any information about a p53-dependence of the observed p21 activation?